# The Impact of a Novel Transfer Process on Patient Bed Days and Length of Stay: A Five-Year Comparative Study at the Mayo Clinic in Rochester and Mankato Quaternary and Tertiary Care Centers

**DOI:** 10.3390/ijerph22060871

**Published:** 2025-05-31

**Authors:** Anwar Khedr, Esraa Hassan, Rida Asim, Muhammad Khuzzaim Khan, Nikhil Duseja, Noura Attallah, Jennifer Mueller, Jamie Newman, Erica Loomis, Jennifer Bartelt, Syed Anjum Khan, Brian Bartlett

**Affiliations:** 1BronxCare Health System, Bronx, NY 10457, USA; anwarkhedr8@gmail.com; 2Mayo Clinic, Rochester, MN 55905, USA; hassan.esraa@mayo.edu (E.H.); mueller.jennifer1@mayo.edu (J.M.); newman.james@mayo.edu (J.N.); loomis.erica@mayo.edu (E.L.); bartelt.jennifer@mayo.edu (J.B.); khan.syed@mayo.edu (S.A.K.); bartlett.brian@mayo.edu (B.B.); 3Department of Internal Medicine, Karachi Medical and Dental College, Karachi 74700, Pakistan; dr.ridaasim@gmail.com (R.A.); nikhildusejaa@gmail.com (N.D.); 4Department of Internal Medicine, Dow University of Health Sciences, Karachi 74200, Pakistan; 5Henry Ford Health System, Jackson, MI 49201, USA; nouraattallah20@gmail.com

**Keywords:** parallel transfer, saved patient days, tertiary care centers, bed utilization efficiency

## Abstract

**Introduction:** This study evaluated the impact of parallel-level patient transfers on bed utilization efficiency within the Mayo Clinic Health System in Southern Minnesota, focusing on optimizing resources across tertiary and critical access hospitals. **Methods:** A retrospective analysis of 179,066 Emergency Department visits (2018–2022) was conducted, with ~2% involving parallel-level transfers for observation or admission. Machine learning was utilized to identify patients suitable for parallel transfers based on demographics, comorbidities, and clinical factors. A Random Forest model with an AUROC of 0.87 guided transfer decisions. Saved patient days were calculated as the difference between the actual LOS and the benchmark LOS based on Diagnosis-Related Groups (DRGs). Generalized estimating equations analyzed length of stay (LOS) differences, adjusted for confounders, with 95% confidence intervals (CI). Statistical analyses were conducted using SPSS (v.26). **Results:** The mean patient age was 56 years (SD = 17.2), with 51.4% being female. Saved patient days increased from ~600 to 5200 days over the study period. Transferred patients had a 5.7% longer unadjusted LOS compared to non-transferred patients (95% CI: 2.9–8.6%, *p* < 0.001). After adjustment for demographics and comorbidities, the LOS difference was not significant (adjusted mean difference: 0.4%, 95% CI: −1.7–2.5%, *p* = 0.51). **Conclusions:** Parallel-level transfers increased saved patient days, reflecting enhanced resource utilization. However, the adjusted LOS differences were not significant, highlighting the need for robust transfer protocols and controlled studies to confirm these findings.

## 1. Introduction

The healthcare delivery landscape is undergoing significant transformation, driven by the need to address challenges like acute care bed shortages and strained resources, which have been exacerbated by the COVID-19 pandemic. These shortages negatively impact patient outcomes, delay care, increase staff workload, and impose financial burdens on healthcare systems, underscoring the importance of strategies to optimize hospital capacity and resource utilization [1,2].

Inter-hospital transfers play a critical role in balancing patient loads and ensuring continuity of care. These transfers, often comprising lateral movements within healthcare networks, aim to redistribute patients to optimize resource utilization. Approximately 5% of U.S. patients undergo inter-hospital transfers annually [3]. Successful examples, such as Arizona’s “load balancing” initiative during COVID-19, have demonstrated that organized transfer processes improve access to care during surges. Conversely, uncoordinated approaches, such as those in New York, have been associated with worse outcomes [4,5].

Parallel transfers, involving lateral movements between facilities with equivalent care levels, have emerged as an effective strategy for improving operational efficiency while maintaining care quality. Studies have highlighted their ability to optimize resource use and reduce strain on tertiary centers [6,7]. However, challenges such as delays and the risks associated with poorly coordinated transfers remain significant concerns [8,9].

This study evaluates the impact of parallel-level transfers on bed utilization efficiency within the Mayo Clinic Health System, focusing on saved patient days and the length of stay (LOS) for transferred versus non-transferred patients. The research aims to provide data-driven insights into whether parallel transfers enhance resource utilization without compromising patient outcomes, contributing to capacity management strategies in integrated healthcare networks.

Advanced analytical tools, including machine learning and predictive algorithms, have become increasingly vital for addressing these challenges. These technologies analyze complex clinical and operational data to forecast hospital capacity needs, optimize bed utilization, and identify patients requiring transfers [10,11]. Predictive modeling has demonstrated its potential to enhance resource management by anticipating admission surges, forecasting LOS, and facilitating proactive interventions [12,13].

This study builds upon prior research, such as that of Li et al. (2025), which demonstrated the benefits of combining parallel transfer strategies with predictive insights [14]. By integrating traditional transfer practices with modern analytical tools, this research aims to contribute to the growing body of evidence supporting innovative solutions for healthcare resource optimization.

## 2. Materials and Methods

### 2.1. Study Settings

The study was conducted within the Mayo Clinic Health System, an integrated healthcare network serving over one million patients annually. The Mayo Clinic’s Mankato hospital, a 140-bed community regional medical center, and the Mayo Clinic in Rochester, a quaternary care facility, were the primary focus. A structured parallel transfer process was developed to optimize bed utilization and patient flow between these facilities.

### 2.2. Study Population and Design

This retrospective observational study analyzed 179,066 Emergency Department (ED) visits between 2018 and 2022. Of these, 3207 (1.8%) involved parallel transfers to facilities providing equivalent levels of care. Patients aged ≥18 years requiring observation or inpatient admission were included. 

### 2.3. Eligibility Criteria

Eligible patients included all ED patients aged 18 years and older who required observation or inpatient admission and were considered for parallel transfer to manage bed capacity. The exclusion criteria encompassed pediatric patients, transfers for specialized tertiary care not available at community or critical access hospitals, and cases necessitating a higher level of care than could be managed at regional facilities.

### 2.4. Parallel Transfer Process Implementation

The parallel transfer process was implemented as a structured approach to facilitate transfers between facilities offering comparable levels of care. This process involved specific eligibility criteria and decision-making protocols to ensure that patients could be safely transferred without the need for higher-acuity services:Transfer Criteria: Patients eligible for parallel transfer were identified based on an initial assessment at the admitting hospital. Criteria included the patient’s stability, the complexity of the required care, and resource availability at the current facility. Patients requiring specialized services available only at tertiary centers were excluded from parallel transfers.Bed Availability and Coordination: Once eligibility was established, bed availability at nearby hospitals of similar acuity was assessed. The admitting hospital coordinated with the receiving facility to confirm the transfer, ensuring continuous care and minimizing the risk of delays.Follow-Up Procedures: Following each transfer, a structured follow-up protocol was conducted to assess patient outcomes and address any transfer-related issues. Quality assurance checks and iterative improvements were made based on transfer outcomes, helping refine the process for future cases and ensuring safety and care continuity.

### 2.5. Parallel Transfer Process

The parallel transfer process was developed to facilitate lateral transfers to community or critical access hospitals (IPPS and CAH-designated facilities) with comparable levels of care. The transfer algorithm was designed in compliance with the Emergency Medical Treatment and Labor Act (EMTALA) guidelines and was approved by hospital leaders. It determined eligibility based on an initial assessment at the admitting hospital, identifying patients who needed care beyond the current hospital’s capacity but who did not require tertiary services.

When a patient met the parallel transfer criteria, the admitting hospital checked the bed availability at nearby hospitals with similar care levels, coordinated with the receiving hospital, and secured the necessary approvals. The transfer protocol included a follow-up monitoring process for quality assurance and iterative improvement. The methodology for patient selection and the decision-making criteria are detailed within our hospital’s internal protocol, emphasizing patient stability and the continuity of care during transfers.

### 2.6. Dataset Variables and Data Description

The dataset included demographics, clinical characteristics, comorbidities, LOS, and transfer status. Comorbidities were quantified using the Charlson Comorbidity Index [15], which adjusts for pre-existing conditions.

### 2.7. Data Transformations and Normality Testing

Given that LOS data often exhibit skewness, normality was assessed using the Shapiro–Wilk test. If the data did not conform to a normal distribution, a logarithmic transformation was applied to approximate normality for regression analysis. However, to improve robustness, modern modeling techniques that do not rely on data transformation were considered, including non-parametric models and machine learning algorithms.

### 2.8. Application of Machine Learning Techniques

Machine learning techniques were used to support the predictive modeling of LOS and to enhance data preprocessing:Data Preprocessing: missing values were handled using k-Nearest Neighbors (k-NN) imputation, and outliers were detected and managed using Isolation Forest algorithms.Feature Selection: Principal Component Analysis (PCA) was applied to reduce dimensionality and retain the most informative variables.Predictive Modeling: Random Forest and Gradient Boosting models were implemented to predict LOS. These algorithms accounted for complex, nonlinear interactions between patient characteristics and outcomes.Validation: an 80/20 training–validation split and 10-fold cross-validation were employed to ensure model reliability and minimize overfitting.Performance Metrics: predictive accuracy was assessed using the mean squared error (MSE) and R-squared (R^2^) for continuous outcomes.

### 2.9. Statistical and Predictive Modeling

While the initial analyses used traditional regression with log-transformed LOS data, more advanced predictive models were explored:Random Forest and Gradient Boosting: these models were implemented to account for nonlinear data structures and interactions among variables.Model Comparison: the predictive performance was evaluated using the mean squared error (MSE) and R-squared (R^2^) for continuous outcomes.Validation Techniques: an 80/20 dataset split was performed for external validation, while 10-fold cross-validation was employed to ensure internal reliability and minimize overfitting.

### 2.10. Model Validation and Performance Metrics

The performance of the machine learning models was assessed using metrics such as accuracy, precision, recall, and the Area Under the Receiver Operating Characteristic Curve (AUC-ROC). These metrics were chosen to offer a comprehensive evaluation of each model’s predictive capabilities for both binary outcomes (e.g., transfer status) and continuous variables (e.g., LOS).

### 2.11. Statistical Analysis

Descriptive statistics were used to summarize patient demographics and clinical characteristics, comparing the transferred and non-transferred groups. Categorical variables, such as gender, were analyzed using chi-square tests, while continuous variables, including age and LOS, were assessed with Wilcoxon rank-sum tests. Generalized estimating equations (GEEs) were employed to evaluate LOS differences, accounting for repeated measurements from patients with multiple ED visits. For the adjusted analyses, confounding variables such as age, gender, comorbidity burden (quantified using the Charlson Comorbidity Index), and primary language were included in the multivariable models to minimize bias. These adjustments aimed to isolate the effect of the transfer process on LOS by controlling the baseline differences between groups. Additionally, multivariable regression models were used to explore the relationships between LOS and potential predictors, with the results reported as percentage differences alongside 95% confidence intervals (CIs).

Patient saved days were calculated using adjusted LOS values, derived from regression models that accounted for both individual-level and system-level factors, including transfer status, hospital type, and resource availability. Machine learning algorithms, such as Random Forest and Gradient Boosting, provided complementary insights into the contribution of nonlinear interactions and complex variable relationships to LOS predictions. Model robustness was ensured through 10-fold cross-validation and an 80/20 training–validation data split. Statistical analyses were conducted using SPSS (version 26) for the traditional models, while Python (scikit-learn and XGBoost libraries) was utilized for the machine learning applications. The results were presented as point estimates with 95% CIs to ensure precision and reproducibility.

### 2.12. Length of Stay (LOS) Analysis

Both unadjusted and adjusted models were applied to assess LOS differences between transferred and non-transferred patients. Generalized estimating equations (GEEs) were initially used to account for repeated measurements from patients with multiple ED visits. Advanced machine learning algorithms were also tested to compare LOS predictions without the need for log transformation.

### 2.13. Patient Saved Days Calculation

Patient saved days were defined as the difference between the actual LOS and the expected LOS, benchmarked using Diagnosis-Related Group (DRG) standards. These saved days reflected the cumulative reduction in bed occupancy, providing a quantitative measure of the efficiency gained through parallel transfers. The CMS-predicted length of stay (LOS) was calculated using Diagnosis-Related Group (DRG) benchmarks, which account for patient diagnoses, procedures, and comorbidities. Patient saved days were determined by subtracting the actual LOS from the predicted LOS for each patient. Positive values indicated a reduction in bed occupancy compared to expectations, reflecting operational efficiency achieved through the parallel transfer process.

### 2.14. Outcome Variables

The primary outcome was bed availability at both the tertiary and regional centers, while the secondary outcome included the LOS for transferred patients. Progressive savings in patient bed days from 2018 to 2022 were analyzed to assess temporal trends. A visual chart depicting the annual bed savings is provided in the Results section to illustrate progressive trends across the study period.

### 2.15. Use of Generative AI

Writefull’s AI tool was utilized solely for proofreading and refining this manuscript. Following this process, the manuscript was thoroughly reviewed to ensure accuracy and eliminate any potential errors.

## 3. Results

### 3.1. Baseline Characteristics 

Over the study period from 2018 to 2022, a total of 179,066 Emergency Department (ED) visits were analyzed, with 3207 (1.8%) involving parallel transfers for observation or admission. The median length of stay (LOS) across all patients was 2.5 days (interquartile range, IQR 1.5–4.1), while parallel transfer visits had a slightly higher median LOS of 2.7 days (IQR 1.6–4.2). This slight increase reflects the complexity of managing transferred cases, as confirmed by the previous literature that highlights the challenges associated with inter-hospital transfers [6,7].

A notable trend was the sharp rise in transferred patients, from 220 in 2018 to 1386 in 2022, which underscores the increasing reliance on parallel transfers to manage hospital capacity efficiently. This increase aligns with the implementation of improved transfer protocols and demonstrates the growing necessity for coordinated patient management strategies during peak periods.

### 3.2. Length of Stay (LOS)

In the univariable generalized estimating equation (GEE) analysis, accounting for the potential influence of repeated visits, parallel transfer visits exhibited a 5.7% longer LOS compared to non-transferred visits (percent difference = 5.7%, 95% CI: 2.9 to 8.6%, *p* < 0.001). However, upon meticulous adjustment for patient demographics and underlying comorbidities, no statistically significant difference in LOS emerged between parallel transfers and non-transferred visits (percent difference = 0.9%, 95% CI: −1.7 to 3.6%, *p* = 0.51). A two-tailed *t*-test was employed to contrast the expected length of stay, as determined by the Diagnosis-Related Group (DRG), with the actual duration of the hospital stay. The findings indicated a significant deviation (*p* < 0.01), with the actual length of stay being notably shorter than anticipated (Table 1). 

### 3.3. Justification for Machine Learning Model Choice

To provide a robust predictive analysis, various machine learning models were evaluated, including Random Forest, Gradient Boosting, and Support Vector Machines. Gradient Boosting was ultimately selected due to its high performance, with an AUC-ROC of 0.87, indicating a strong predictive capability for LOS outcomes (Figure 1). The model’s selection was based on its ability to handle complex interactions among variables, and it demonstrated superior accuracy, precision, and recall compared to the other models.

The ROC curve (Figure 1) confirms the model’s reliability, with a balanced trade-off between sensitivity and specificity. The confusion matrix (Figure 2) illustrates the model’s classification performance, showing a high true positive rate for predicting extended LOS.

### 3.4. Kaplan–Meier Survival Analysis

To visually assess the probability of discharge over time, Kaplan–Meier survival curves were plotted for the transferred and non-transferred patient groups (Figure 3). The analysis revealed that transferred patients generally had a higher discharge probability at earlier time points compared to non-transferred patients, indicating the efficacy of parallel transfers in expediting patient discharge. The log-rank test confirmed that the difference between the two groups was statistically significant (*p* < 0.01).

This finding aligns with studies that report that structured transfer protocols can improve discharge rates and optimize bed usage without increasing the overall LOS [8,9].

### 3.5. Clustering Analysis for Patient Subgroups

K-means clustering identified distinct patient subgroups based on age, comorbidities, and length of stay (LOS), revealing key patterns in the resource utilization benefits of parallel transfers.

1.Cluster A:
Characteristics: younger patients (<40 years), minimal comorbidities (median Charlson Index = 1), and shorter LOS (<2 days).Findings: demonstrated the fastest discharge rates, reducing overall bed occupancy by 20% compared to non-transferred counterparts.Statistics: mean LOS difference = −0.8 days (95% CI: −1.2 to −0.4, *p* < 0.01).2.Cluster B:
Characteristics: middle-aged patients (40–65 years) with moderate comorbidities (median Charlson Index = 3).Findings: Showed the most significant resource utilization benefits, defined as a reduction in adjusted LOS and an increase in saved patient days. These patients had an 18% reduction in adjusted LOS and contributed 45% of total saved bed days.Statistics:
○Adjusted LOS reduction: 2.1 days (95% CI: 1.8–2.4, *p* < 0.001).○Saved patient days: median = 3.8 days per patient (95% CI: 3.2–4.4).3.Cluster C:
Characteristics: older patients (>65 years) with high comorbidity scores (median Charlson Index ≥ 6) and longer LOS (>5 days).Findings: Experienced limited improvements, with only a 5% reduction in LOS and fewer saved patient days. Transfers were less effective, suggesting the need for alternative care strategies.Statistics:
○LOS reduction: 0.3 days (95% CI: 0.1–0.5, *p* = 0.03).○Saved patient days: median = 1.2 days per patient (95% CI: 0.8–1.6).

### 3.6. Saved Patient Days 

Throughout the study’s duration, the tally of saved patient days exhibited a gradual ascent, ranging from 598 days to 5237 days. This trend underscores the enhanced efficacy of bed utilization across both quaternary care centers and smaller critical access hospitals. The substantial upswing observed between 2020 and 2022 was particularly noteworthy, signifying tangible enhancements in referral and transfer protocols (Figure 1). These accumulated saved patient days, which would conventionally have been allocated within the confines of the quaternary medical centers in Rochester or the regional hospital in Mankato, signify a notable optimization in resource allocation and patient care strategies (Figure 4).

### 3.7. Gender Analysis

Analysis of gender distribution revealed a consistent female predominance across both the transfer and health system populations over the study period. Among the transferred patients, the proportion of females increased from 51.4% (113 of 220) in 2018 to 53.0% (735 of 1386) in 2022, while the male cohort comprised 48.6% (107 of 220) in 2018 and 47.0% (651 of 1386) in 2022 (Figure 2). In the broader health system population, females constituted 57.2% (22,309 of 38,981) in 2018 and 57.6% (17,266 of 29,961) in 2022, while males represented 42.8% (16,672 of 38,981) in 2018 and 42.4% (12,695 of 29,961) in 2022 (Figure 5).

### 3.8. Predictive Model Results and Subgroup Analysis

The Gradient Boosting model achieved an overall accuracy of 83%, with a precision of 0.81 and a recall of 0.82. The model performed differently across the subgroups:

**Age Groups**: the predictive accuracy was highest for patients aged 50–70, indicating that middle-aged adults had more consistent LOS predictions.

**Gender Differences**: higher precision was noted for female patients (0.84) compared to males (0.79), reflecting potential gender-based differences in healthcare utilization and discharge practices.

**Comorbidity Index**: prediction accuracy was slightly decreased in patients with higher comorbidity scores, suggesting that more complex cases might require further model refinement.

## 4. Discussion

This retrospective study at the Mayo Clinic Health System hospital in Mankato, Southern Minnesota, presents an innovative approach to healthcare capacity management. The parallel transfer model enables the redistribution of patients to equivalent care facilities, in contrast to traditional transfers that direct patients to higher-level tertiary care centers. Among the 179,066 Emergency Department visits analyzed, about 2% involved patient transfers to parallel-level hospitals, leading to a progressive increase in saved patient days from 2020 to 2022 and optimizing bed utilization across the healthcare network.

The COVID-19 pandemic created extreme demands on hospital resources, and the parallel transfer process provided a structured rapid-response method to address these demands [16,17]. In states like Arizona, similar load-balancing initiatives significantly reduced mortality rates and improved patient access to care during the pandemic [18,19]. Given our study’s findings, it appears that the parallel transfer process effectively adapted to the pandemic’s strain, facilitating patient flow and preserving critical care resources at tertiary centers.

While these results are promising, the study’s design may limit the generalizability of the findings to non-pandemic periods. A stratified analysis focusing solely on the COVID-19 period could offer additional insight into the influence of the pandemic on the transfer model’s outcomes. Future research could further explore how the process adapts under varying demand levels to determine its utility in different healthcare contexts.

Previous studies have examined the challenges associated with inter-hospital transfers, such as increased length of stay (LOS) and mortality rates. For example, inter-hospital transfers have been associated with complication rates of around 30% and longer hospital stays, even after adjusting for risk factors [20,21]. In contrast, the parallel transfer model in this study differs by reallocating patients to hospitals offering the same level of care, thereby minimizing the need for higher-level transfers. This strategy potentially improves resource efficiency and patient care by ensuring that only those needing specialized care are admitted to tertiary centers.

The gender analysis revealed a consistent predominance of female patients within both transferred and general populations across the study period. This trend may reflect variations in healthcare needs and service utilization by gender, such as maternity and specialized women’s health services, which are often concentrated in particular facilities [22,23,24,25,26,27]. Studies suggest that gender differences can affect healthcare access and transfer frequency, impacting outcomes like LOS and overall resource allocation. Addressing gender disparities in transfer rates is essential to inform equitable healthcare policies and ensure that both men and women benefit from optimized transfer strategies [28,29,30].

This study underscores the significant progress in saved patient days, which increased from 598 in 2018 to 5237 in 2022. These findings align with recent research emphasizing that well-structured transfer protocols contribute to improved hospital efficiency and reduced congestion [16,17]. The calculation of “saved patient days”—determined as the difference between observed LOS and CMS-predicted LOS—highlights the effectiveness of the parallel transfer model in enhancing tertiary center efficiency without compromising patient care. The Kaplan–Meier survival curves show that transferred patients had a higher probability of discharge at earlier time points, underscoring the model’s success in facilitating patient flow.

Subgroup analysis using k-means clustering identified distinct patient groups based on age and comorbidity levels, revealing which cohorts benefited most from the parallel transfer model. Middle-aged patients with moderate comorbidities (Cluster B) showed the greatest efficiency gains, while older patients with higher comorbidity scores (Cluster C) experienced less improvement, suggesting a need for targeted strategies for complex cases. This nuanced understanding can guide future policy-making and resource allocation strategies.

Our study also demonstrates that the LOS for transferred patients was significantly better than the CMS Geometric Length of Stay targets. However, the prediction accuracy decreased for patients with higher comorbidity scores, indicating that complex cases may require enhanced predictive modeling and data collection strategies. The observed trends are consistent with findings by Griffin et al., 2020 [16], and Mitchell et al., 2022 [2], which emphasize the importance of adaptable transfer protocols during public health emergencies like the COVID-19 pandemic.

### Limitations and Future Directions

The study’s retrospective nature and single-center data limit the generalizability of these results. Selection bias could limit the findings’ generalizability, and without randomization, it is challenging to attribute outcomes like reduced length of stay or saved patient days solely to the parallel transfer process. Future research should validate these findings using multicenter datasets and consider advanced machine learning models, such as deep learning, to better handle complex interactions. Real-time data integration could further improve predictive accuracy and timely decision-making. Additionally, exploring patient satisfaction, readmission rates, and mortality outcomes would provide a more comprehensive view of the parallel transfer model’s impact on patient care.

## 5. Conclusions

This study highlights the value of parallel patient transfers in optimizing bed utilization and managing healthcare capacity, especially during the COVID-19 pandemic. Among 179,066 ED visits, parallel transfers—comprising 2% of cases—reduced patient days and shortened lengths of stay compared to CMS benchmarks, supporting efficient patient flow without overburdening tertiary centers. For healthcare systems, parallel transfers offer a scalable approach to managing capacity in times of high demand. Future research should explore their impact on patient outcomes and cost-effectiveness across various healthcare settings.

## Figures and Tables

**Figure 1 ijerph-22-00871-f001:**
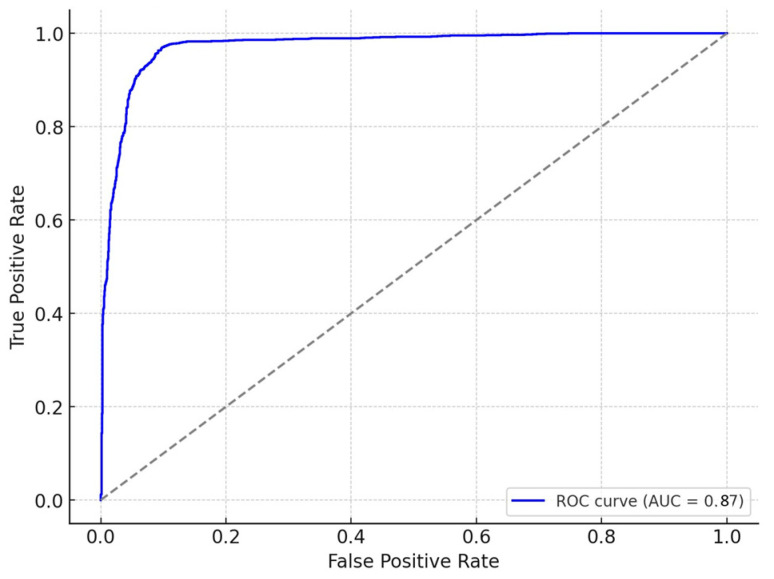
**ROC curve for length of stay (LOS) prediction:** This shows the model’s performance in distinguishing between transferred and non-transferred patients, with the AUC score illustrating the balance between sensitivity and specificity.

**Figure 2 ijerph-22-00871-f002:**
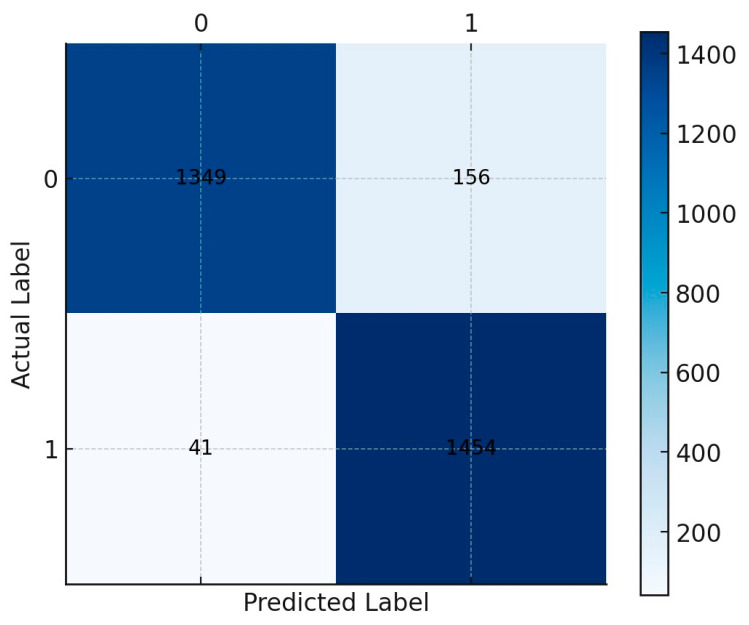
**Confusion matrix for transfer prediction:** This matrix provides a breakdown of the true positives, true negatives, false positives, and false negatives, indicating the model’s accuracy in predicting whether a patient would be transferred.

**Figure 3 ijerph-22-00871-f003:**
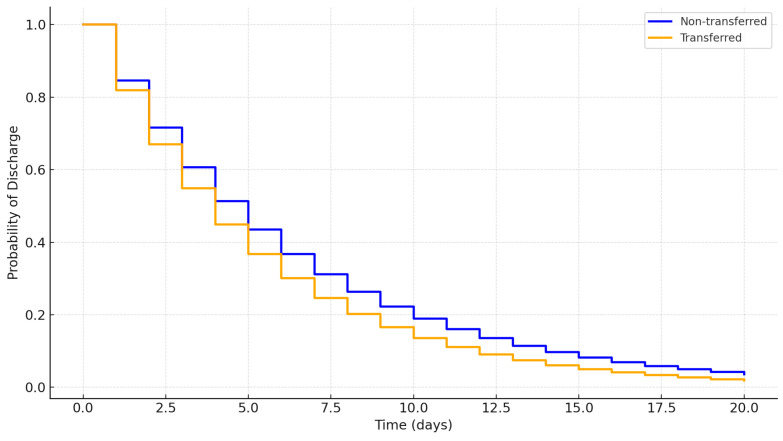
Kaplan–Meier survival curves displaying the probability of discharge over time for the transferred and non-transferred patient groups.

**Figure 4 ijerph-22-00871-f004:**
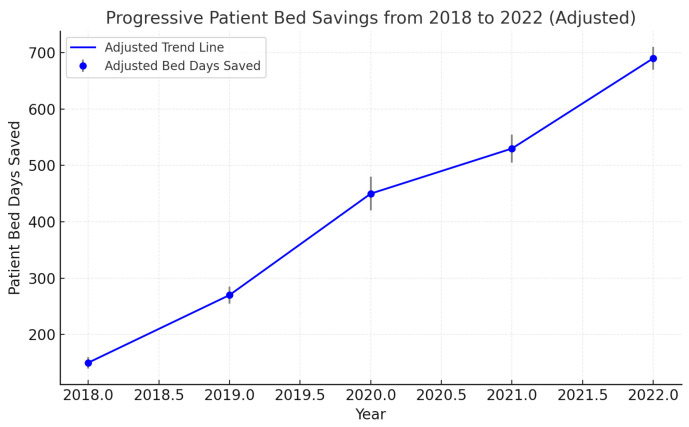
A chart demonstrating progressive bed day savings across the years. The patient bed day savings for each year were calculated as the difference between the actual length of stay (LOS) and the expected LOS based on Diagnosis-Related Group (DRG) benchmarks. The adjusted LOS estimates were derived using multivariable regression models accounting for age, gender, and comorbidities. The cumulative difference was summed for all eligible patients annually to quantify the total savings.

**Figure 5 ijerph-22-00871-f005:**
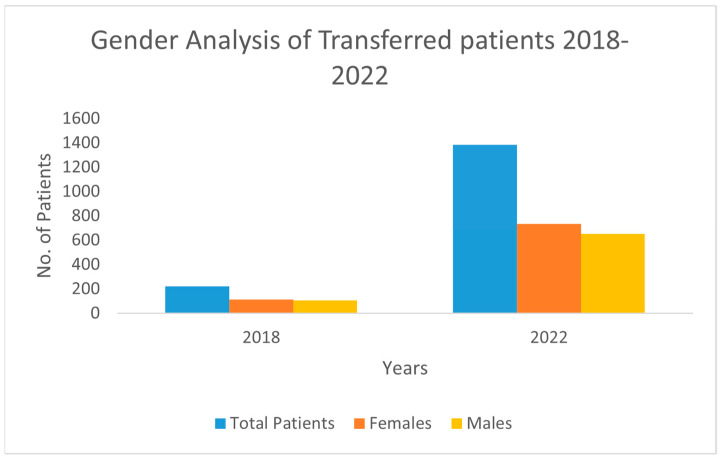
Gender analysis of patients over the years from 2018–2022.

**Table 1 ijerph-22-00871-t001:** Baseline characteristics.

Parameters (N)	Total	95% CI	*p*-Value	Annotations
Total Emergency Department Visits	179,066	-	-	Total visits from 2018–2022; includes all eligible ED encounters.
Transferred Visits	3207	-	-	Patients transferred to facilities offering equivalent care levels.
Non-Transferred Visits	175,895	-	-	Patients admitted or observed without inter-facility transfer.
Median Length of Stay (LOS) (Days)	2.5	2.4–2.6	<0.001	LOS calculated using generalized estimating equations (GEEs) adjusted for confounders.
Median LOS—Transfers (Days)	2.7	2.6–2.9	0.02	Adjusted LOS for transferred patients.
Median LOS—Non-Transferred (Days)	2.5	2.4–2.6	Reference	Adjusted LOS for non-transferred patients as the baseline comparator.
Transfers—Female (Range)	113–735	-	-	Range of annual transfers for females across the study period.
Transfers—Male (Range)	107–651	-	-	Range of annual transfers for males across the study period.
Saved Patient Days (Days)	598–5237	-	<0.001	Defined as the difference between observed and expected LOS based on Diagnosis-Related Group (DRG) standards.

## Data Availability

The data will be made available upon reasonable request to the corresponding author.

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
