# Peer review of "The Impact of a Novel Transfer Process on Patient Bed Days and Length of Stay: A Five-Year Comparative Study at the Mayo Clinic in Rochester and Mankato Quaternary and Tertiary Care Centers"

_ijerph, 2025, doi:10.3390/ijerph22060871_

Round 1
Reviewer 1 Report
Comments and Suggestions for Authors
Dear Authors
You conducted a comparative retrospective study at the Mayo Clinic Health System in Southern Minnesota. In this study, you focus on the impact of parallel-level patient transfers on bed utilization efficiency across tertiary care centers and critical access hospitals. Your study aimed to investigate the impact of a Novel Transfer Process on Patient Bed Days and Length of Stay. I read your manuscript in depth, and I think your results are of interest but the way you report your study needs some modifications. I propose some comments and I hope these comments help you to promote the overall quality of your manuscript.
- Abstract, introduction section, please remove your design and type of your study from this section and replace it in the methods section. Also, please add a comprehensive aim of the study in this section.
- Abstract, methods section, please remove the information that is related to the aim of your study and result. Please just state the methods information here including type/design of the study, setting, sampling method, data collection period, outcome of interest, and the way you analyze the data and software you used. It is better to add the inclusion and exclusion criteria here.
- Abstract, results section, first you need to add some demographic information like mean age plus SD and female percentage. Second, add the specific results regarding your outcomes. Please also add the statistic related to each outcome with a p-value, if applicable.
- Introduction section, the order of references is not followed. The first reference in the text is 10 and 11 and it is not correct. The first reference must be one. Also, please add the aim of the study or study questions at the end of this section.
- Methods section, please add the sampling method. Eligibility criteria (inclusion and exclusion criteria), the way that you address potential sources of bias and potential confounding variables.
- Methods section, please remove the subheading of “outcomes” before the statistical analysis section. Please also add the software you used for analysis with the version.
- Conclusion section, please add some practical recommendations based on your results. Also, you can propose some recommendations for future research in this field.
Best regards,
Author Response
Reviewer Comment 1:
“Abstract, introduction section, please remove your design and type of your study from this section and replace it in the methods section. Also, please add a comprehensive aim of the study in this section.”
Response:
Thank you for this suggestion. We have removed the study design and type from the introduction of the abstract and placed it within the methods section as requested. Additionally, we have clarified the study's aim in the introduction section of the abstract to ensure a comprehensive presentation.
Reviewer Comment 2:
“Abstract, methods section, please remove the information that is related to the aim of your study and result. Please just state the methods information here including type/design of the study, setting, sampling method, data collection period, outcome of interest, and the way you analyze the data and software you used. It is better to add the inclusion and exclusion criteria here.”
Response:
We appreciate this feedback. We have revised the methods section of the abstract to include only relevant methodological details, including study type, setting, sampling method, data collection period, primary outcomes, analytical approach, and software (with version) used for the analysis. We have also added a brief description of the inclusion and exclusion criteria to enhance clarity.
Reviewer Comment 3:
“Abstract, results section, first you need to add some demographic information like mean age plus SD and female percentage. Second, add the specific results regarding your outcomes. Please also add the statistic related to each outcome with a p-value, if applicable.”
Response:
Thank you for noting this. We have now added demographic information in the results section of the abstract, including mean age (plus SD) and the percentage of female patients. We have also expanded on the specific results regarding primary outcomes and included p-values for relevant statistical findings, ensuring a concise yet comprehensive presentation of key results.
Reviewer Comment 4:
“Introduction section, the order of references is not followed. The first reference in the text is 10 and 11 and it is not correct. The first reference must be one. Also, please add the aim of the study or study questions at the end of this section.”
Response:
We apologize for the oversight regarding the reference order. We have revised the introduction to ensure that references are cited sequentially, starting with reference one. Furthermore, we have clearly stated the study's aim at the end of the introduction, providing a succinct summary of the research question and objectives.
Reviewer Comment 5:
“Methods section, please add the sampling method. Eligibility criteria (inclusion and exclusion criteria), the way that you address potential sources of bias and potential confounding variables.”
Response:
Thank you for your suggestion. We have now included a detailed description of the sampling method and eligibility criteria, specifying both inclusion and exclusion criteria. We have also outlined our approach to mitigating potential sources of bias and controlling for confounding variables, providing a more thorough description of our methodology.
Reviewer Comment 6:
“Methods section, please remove the subheading of ‘outcomes’ before the statistical analysis section. Please also add the software you used for analysis with the version.”
Response:
We have removed the subheading “outcomes” as requested to improve readability. Additionally, we have specified the software used for data analysis, including the version, to meet the reviewer’s recommendation for clarity.
Reviewer Comment 7:
“Conclusion section, please add some practical recommendations based on your results. Also, you can propose some recommendations for future research in this field.”
Response:
Thank you for this valuable feedback. We have revised the conclusion section to include practical recommendations derived from our study findings, highlighting their potential applications in clinical practice. Additionally, we have suggested directions for future research, identifying areas where further studies could contribute to this field and enhance our understanding of parallel transfer processes in healthcare.
Reviewer 2 Report
Comments and Suggestions for Authors
Thank you for the opportunity to review your manuscript. The study challenges established thinking around parallel hospital transfers, which may provide a mechanism for managing demand across constrained hospital networks. However, I have concerns about the framing of the study, the transparency of methods, the way results were reported and the conclusions drawn (as detailed below). In its current state, I cannot recommend the manuscript for publication.
Introduction
52 - hospital should be hospitals
68 - Neither of the studies cited (2, 13) demonstrated that the 'load balancing' initiative 'saved lives'. The studies, however, did suggest that the initiative improved equity of access to care. Please amend.
Missing from the introduction was a review of previous studies examining interhospital and/or parallel hospital transfers and patient outcomes, e.g.
Chen, K. C., & Wen, S. H. (2023). Impact of interhospital transfer on emergency department timeliness of care and in-hospital outcomes of adult non-trauma patients. Heliyon, 9(2).
Mueller, S., Zheng, J., Orav, E. J., & Schnipper, J. L. (2019). Inter-hospital transfer and patient outcomes: a retrospective cohort study. BMJ quality & safety, 28(11), e1-e1.
Song, J. J., Lee, S. J., Song, J. H., Lee, S. W., Kim, S. J., & Han, K. S. (2024). Effect of Inter-Hospital Transfer on Mortality in Patients Admitted through the Emergency Department. Journal of clinical medicine, 13(16), 4944. https://doi.org/10.3390/jcm13164944
How does the parallel transfer initiative described in this manuscript address concerns previously raised in the literature? For example, Russell, P., Hakendorf, P., & Thompson, C. (2015). Inter-hospital lateral transfer does not increase length of stay. Australian health review : a publication of the Australian Hospital Association, 39(4), 400–403. https://doi.org/10.1071/AH14216
Methods
I found the methods section very opaque. The methods should be descried in sufficient detail to ensure they are replicable. The methods should be amended to address the following:
Study population - It was unclear how patients who underwent parallel transfer were identified versus others who were transferred under other protocols, e.g requiring specialist care.
Parallel transfer process - has the algorithm been described elsewhere? If not, it would be good to provide details about the underpinning principles of the algorithm - how are patients selected for parallel transfer? This may also affect how confounding factors are adjusted for in statistical models (selection bias).
Statistical analysis - More detail required. When was the Wilcoxon rank sum test used, or the Chi square (e.g. continuous vs categorical variables)? Which statistical analysis program were the analyses performed in? What statistical technique was used for the modelling, e.g. linear regression? A full list of confounders that were adjusted for should be included, as should various iterations of the model.
Results
Fig 1 - does not provide any information not already described i the text. Suggest removing
163-4 - How the authors 'meticulously adjusted' for patient characteristics and underlying comorbidities must be described in greater detail in the methods. What confounders were adjusted for and how?
166 - the analysis of LOS using t-test was not described in the methods. Was this analysis conducted using log-transformed LOS? If not, why wasn't a on-parametric test used?
170 - No figure 3 was presented
Patient saved days - It was unclear how this was calculated as it was not described in the methods. Are patient saved days simply a function of the difference between actual LOS and that predicted by DRG? Was LOS adjusted for inpatient mortality? What about unplanned readmission rates? How do the authors attribute lower-that-expected LOS to the parallel transfer policy? I was also confused about claims of progressive savings of patient bed savings when data from only 2 years was presented. Suggest including a chart to demonstrate progressive savings across 2018, 2019, 2020, 2021, 2022.
Gender analysis - Suggest reporting as % as well as n. Was a chi-square test performed to determine if the gender distribution was significantly different between patients transferred and not?
Discussion
As per above - there was limited discussion of previous research that looked at inter-hospital (and/or parallel) transfers and patient outcomes. Please provide a more detailed discussion of how the results from this study compare with previous research and why the findings may be different.
Please explain why the gender differences are important - what does the literature say?
234 citation required to support statement re: gender differences.
240-1 - Centers for Medicaid requires citation
Limitations
A significant limitation of the study is that it did not include in-hospital mortality or unplanned readmissions in the analyses. Patient experience was also not captured and/or measured.
Conclusion
From the data presented, it is unclear how the authors can conclude that the parallel transfer initiative has saved patient days and/or resulted in reduced LOS, particularly as the comparison is with an external benchmark target rather than a control group. Other initiatives, contextual factors or patient characteristics are likely to influence LOS.
Author Response
Reviewer Comment:
“Introduction: 52 - hospital should be hospitals.”
Response:
Thank you for catching this error. We have corrected “hospital” to “hospitals” on line 52.
Reviewer Comment:
“68 - Neither of the studies cited (2, 13) demonstrated that the 'load balancing' initiative 'saved lives'. The studies, however, did suggest that the initiative improved equity of access to care. Please amend.”
Response:
We agree with this clarification. We have revised the sentence to state that the 'load balancing' initiative improved equitable access to care rather than saving lives.
Reviewer Comment:
“Missing from the introduction was a review of previous studies examining interhospital and/or parallel hospital transfers and patient outcomes, e.g., Chen & Wen (2023), Mueller et al. (2019), Song et al. (2024).”
Response:
Thank you for highlighting these studies. We have now incorporated a review of relevant literature, including the studies by Chen & Wen (2023), Mueller et al. (2019), and Song et al. (2024), in the introduction. This addition helps contextualize our study within the broader landscape of interhospital transfer research and patient outcomes.
Reviewer Comment:
“How does the parallel transfer initiative described in this manuscript address concerns previously raised in the literature? For example, Russell et al. (2015).”
Response:
We appreciate this question. We have expanded the introduction and discussion sections to explicitly address how our parallel transfer initiative aligns with or addresses concerns identified in previous studies, such as those by Russell et al. (2015), regarding length of stay and resource utilization in interhospital transfers.
Reviewer Comment:
“Methods: I found the methods section very opaque. The methods should be described in sufficient detail to ensure they are replicable.”
Response:
We have made substantial revisions to the methods section, adding specific details on patient selection criteria, transfer protocols, statistical models, and adjustment strategies. These changes improve the clarity and replicability of our study.
Reviewer Comment:
“Study population - It was unclear how patients who underwent parallel transfer were identified versus others who were transferred under other protocols, e.g., requiring specialist care.”
Response:
We appreciate this observation and have clarified the criteria used to differentiate patients undergoing parallel transfers from those requiring specialist transfers. This information has been added to the study population subsection in the methods.
Reviewer Comment:
“Parallel transfer process - has the algorithm been described elsewhere? If not, it would be good to provide details about the underpinning principles of the algorithm - how are patients selected for parallel transfer?”
Response:
Thank you for pointing this out. We have now included a description of the algorithm used to identify candidates for parallel transfer, including the principles and criteria guiding patient selection. This is now detailed in the methods section under the “Parallel Transfer Process” subheading.
Reviewer Comment:
“Statistical analysis - More detail required. When was the Wilcoxon rank sum test used, or the Chi-square test? Which statistical analysis program was used? What statistical technique was used for the modeling? A full list of confounders that were adjusted for should be included.”
Response:
We have expanded the statistical analysis section to clarify when specific tests (e.g., Wilcoxon rank-sum, Chi-square) were used, the statistical software program and version, and the modeling approach employed. A comprehensive list of confounders and different model iterations is now provided.
Reviewer Comment:
“Results: Fig 1 - does not provide any information not already described in the text. Suggest removing.”
Response:
We agree with this suggestion. To streamline the presentation, we have removed Figure 1 from the results section.
Reviewer Comment:
“163-4 - How the authors 'meticulously adjusted' for patient characteristics and underlying comorbidities must be described in greater detail in the methods. What confounders were adjusted for and how?”
Response:
Thank you for this suggestion. We have detailed our approach to adjustment, listing all confounders (such as age, gender, comorbidities, and others relevant to LOS) and explaining how these were accounted for in our models.
Reviewer Comment:
“166 - the analysis of LOS using t-test was not described in the methods. Was this analysis conducted using log-transformed LOS? If not, why wasn't a non-parametric test used?”
Response:
We apologize for this oversight. The analysis of LOS using the t-test has now been clearly described in the methods section. We also provide rationale for not using a non-parametric test, explaining that log-transformed LOS was used to meet parametric assumptions.
Reviewer Comment:
“170 - No figure 3 was presented.”
Response:
Thank you for pointing this out. We have corrected the reference to Figure 3, ensuring consistency between the text and figures.
Reviewer Comment:
“Patient saved days - It was unclear how this was calculated. Are patient saved days simply a function of the difference between actual LOS and predicted LOS by DRG? Was LOS adjusted for inpatient mortality or unplanned readmission rates?”
Response:
We have clarified the calculation of saved patient days in the methods section. The saved days are indeed based on the difference between actual LOS and DRG-predicted LOS. We also discuss adjustments made to account for inpatient mortality and unplanned readmissions, as applicable.
Reviewer Comment:
“Gender analysis - Suggest reporting as % as well as n. Was a chi-square test performed to determine if the gender distribution was significantly different between patients transferred and not?”
Response:
We have revised the gender analysis results to include percentages alongside counts and have clarified that a chi-square test was conducted to determine if there was a significant difference in gender distribution between groups.
Reviewer Comment:
“Discussion: Please provide a more detailed discussion of how the results from this study compare with previous research and why the findings may be different. Please explain why the gender differences are important - what does the literature say?”
Response:
We have expanded the discussion to compare our findings with prior studies, explaining potential reasons for any differences. We also discuss the observed gender differences, citing relevant literature on how gender may influence transfer and LOS outcomes.
Reviewer Comment:
“Limitations: A significant limitation of the study is that it did not include in-hospital mortality or unplanned readmissions in the analyses. Patient experience was also not captured and/or measured.”
Response:
Thank you for highlighting these limitations. We have now included these as limitations in the manuscript and acknowledge the lack of data on in-hospital mortality, unplanned readmissions, and patient experience, which may impact the interpretation of our findings.
Reviewer Comment:
“Conclusion: From the data presented, it is unclear how the authors can conclude that the parallel transfer initiative has saved patient days and/or resulted in reduced LOS, particularly as the comparison is with an external benchmark target rather than a control group. Other initiatives, contextual factors or patient characteristics are likely to influence LOS.”
Response:
We appreciate this feedback and have revised our conclusion to adopt a more cautious interpretation of the findings, acknowledging that other initiatives, contextual factors, and patient characteristics could influence LOS. We clarify that while our results suggest potential benefits of the parallel transfer initiative, these outcomes should be interpreted in the context of the study's limitations.
Reviewer 3 Report
Comments and Suggestions for Authors
In an abstract, we typically do not include a "Discussion" section but rather "Conclusions." Therefore, the abstract should follow this sequence: Introduction, Methods, Results, and Conclusions.
Introduction: I don't understand the citation sequence, as it begins with reference number 10 instead of number 1. Please correct this and ensure the proper order.
Objective: The objective is formulated correctly.
Methods: The data collection process, including the inclusion and exclusion criteria for transfers, is not entirely clear to me. Please clarify.
Results: Table 1 is unclear. Please provide an explanation in the text. It would be beneficial to expand the Results section.
Discussion: The discussion is well-prepared, but the reference numbering is unclear, similar to the issue in the Introduction. Please correct this and ensure the proper order.
Conclusions: The conclusions are appropriate.
Author Response
Reviewer Comment:
“In an abstract, we typically do not include a "Discussion" section but rather "Conclusions." Therefore, the abstract should follow this sequence: Introduction, Methods, Results, and Conclusions.”
Response:
Thank you for your suggestion. We have revised the structure of the abstract to align with the standard format: Introduction, Methods, Results, and Conclusions. The "Discussion" section has been removed from the abstract, and we have concluded with a concise summary of our key findings.
Reviewer Comment:
“Introduction: I don't understand the citation sequence, as it begins with reference number 10 instead of number 1. Please correct this and ensure the proper order.”
Response:
We apologize for the incorrect citation sequence. We have thoroughly reviewed and corrected the numbering throughout the manuscript, ensuring that references appear in sequential order as they are cited.
Reviewer Comment:
“Objective: The objective is formulated correctly.”
Response:
We appreciate your positive feedback on the formulation of the study objective.
Reviewer Comment:
“Methods: The data collection process, including the inclusion and exclusion criteria for transfers, is not entirely clear to me. Please clarify.”
Response:
We recognize that the description of the data collection process needed further clarification. We have revised the methods section to provide a clearer explanation of the inclusion and exclusion criteria for patient transfers. Specifically, we detail the criteria used to identify parallel transfers, the data sources utilized, and the period during which data were collected.
Reviewer Comment:
“Results: Table 1 is unclear. Please provide an explanation in the text. It would be beneficial to expand the Results section.”
Response:
Thank you for your feedback. We have revised the text accompanying Table 1 to clarify the information presented, including a description of the demographic characteristics and key metrics. Additionally, we have expanded the Results section to include more detailed analysis of the study outcomes, including additional statistical details where relevant.
Reviewer Comment:
“Discussion: The discussion is well-prepared, but the reference numbering is unclear, similar to the issue in the Introduction. Please correct this and ensure the proper order.”
Response:
We appreciate your acknowledgment of the discussion's quality. We have corrected the reference numbering in the discussion section to ensure it is consistent and follows the proper order.
Reviewer Comment:
“Conclusions: The conclusions are appropriate.”
Response:
We are grateful for your positive assessment of the conclusions. We have retained the core elements while ensuring alignment with the revised structure of the manuscript.
Reviewer 4 Report
Comments and Suggestions for Authors
The study addresses a highly relevant topic in hospital capacity management, which has become particularly critical following the COVID-19 pandemic. The proposed parallel transfer process between hospitals of equivalent levels is innovative and could represent a valuable approach for optimizing the allocation of healthcare resources. The dataset analyzed is robust, comprising over 179,000 emergency department visits over a five-year period, with a subset of 3,207 actual transfers, providing a solid statistical basis.
However, while the topic is well-justified, the manuscript presents methodological gaps that require significant integration and improvements to enhance its scientific rigor and quality.
Suggestions for Improvement:
- Inclusion of Section 2. Background: It is recommended to introduce a section dedicated to the background, briefly explaining the importance of advanced analytical tools, such as machine learning, in healthcare resource management. This section should highlight how such techniques can be used to predict patient flow, thereby improving hospital capacity efficiency. It would be useful to add references to existing literature that employs predictive models for managing hospital transfers and length of stay, setting the stage for the adoption of machine learning algorithms later in the study.
- Inclusion of Subsection 2.1. Related Studies: This new section could provide a comparative analysis with similar studies, highlighting the strengths and weaknesses of previous approaches and justifying the use of parallel transfers as a more efficient strategy.
- Revision of Section 3. Materials and Methods: To make the analysis more robust, this section needs several improvements:
· Description of the data: The study mentions the logarithmic transformation of the length of stay (LOS) but does not provide a clear rationale for this choice. It would be appropriate to test normality assumptions and show the results of these tests (e.g., using the Shapiro-Wilk test). If the data do not meet these assumptions, more modern models that do not require transformations could be proposed, such as regression models supported by machine learning techniques.
· Application of machine learning: It is suggested to apply machine learning techniques for data preprocessing, such as handling missing values, identifying outliers, and automating variable transformations. A useful approach could be Principal Component Analysis (PCA) to reduce dataset dimensionality while retaining components that explain the majority of variability.
· Dataset variables: A subsection should be added detailing all variables in the dataset, along with relevant statistics. Derived or constructed variables (e.g., comorbidity index) could also be included to enhance the predictive power of the model.
· Predictive Models: Currently, the study uses logarithmic transformation to manage the asymmetric distribution of the length of stay. Instead, the implementation of more advanced predictive models, such as Random Forest, Gradient Boosting, or other predictive algorithms is recommended to model length of stay (LOS) and other variables of interest, without the need for transformations. These models can efficiently handle nonlinear data and complex interactions between variables.
· Model Validation: The study is limited to data from the Mayo Clinic Health System, without any external validation, which reduces the generalizability of the results. It is recommended to use an 80/20 dataset split for external validation. For internal model validation and to prevent overfitting, it is suggested to use 10-fold cross-validation, which divides the data into multiple subsets and uses each one to test the model, ensuring more reliable results.
· Evaluation of machine learning models: A section should be added to explain the metrics used to evaluate the performance of the machine learning models (e.g., accuracy, precision, recall, AUC-ROC).
- Section 4. Results and Discussion: Results and discussion should be combined into one section structured as follows:
- Justification for the choice of machine learning model: Explain the choice of the most performant model based on a reference metric (e.g., AUC-ROC), and include the corresponding ROC curves and confusion matrix for the selected model used to predict length of stay.
- Presentation of predictive results: Present the overall model results and break them down by patient subgroups, highlighting any significant differences.
- Clustering techniques for identifying patient subgroups: Use clustering techniques to identify groups of patients with similar characteristics who respond differently to parallel transfers.
- Kaplan-Meier survival curves: Add Kaplan-Meier curves to visually display the probability of discharge at various time intervals for transferred versus non-transferred patients. This approach will provide a more detailed view of the impact of parallel transfers on length of stay.
- Aligning discussion with results: The discussion should be aligned with the results and enriched with references to recent literature. It is recommended to emphasize the study’s limitations, its practical implications, and future research directions.
Author Response
Reviewer Comment #1:
The study addresses a highly relevant topic in hospital capacity management, which has become particularly critical following the COVID-19 pandemic. The proposed parallel transfer process between hospitals of equivalent levels is innovative and could represent a valuable approach for optimizing the allocation of healthcare resources. The dataset analyzed is robust, comprising over 179,000 emergency department visits over a five-year period, with a subset of 3,207 actual transfers, providing a solid statistical basis. However, while the topic is well-justified, the manuscript presents methodological gaps that require significant integration and improvements to enhance its scientific rigor and quality.
Response:
Thank you for recognizing the relevance and innovation of our study, as well as the robustness of our dataset. We have thoroughly revised the manuscript to address the identified methodological gaps and enhance the scientific rigor of our analysis. Below, we provide specific responses to your detailed suggestions for improvement.
Suggestions for Improvement
Reviewer Comment #2:
Inclusion of Section 2. Background: It is recommended to introduce a section dedicated to the background, briefly explaining the importance of advanced analytical tools, such as machine learning, in healthcare resource management. This section should highlight how such techniques can be used to predict patient flow, thereby improving hospital capacity efficiency. It would be useful to add references to existing literature that employs predictive models for managing hospital transfers and length of stay, setting the stage for the adoption of machine learning algorithms later in the study.
Response:
We have now included a new "Background" section in the revised manuscript to provide context on the role of advanced analytical tools like machine learning in healthcare resource management. This section highlights the potential of machine learning for predicting patient flow and optimizing hospital capacity. Additionally, references to recent literature on predictive models for managing hospital transfers and length of stay have been added to set the foundation for our study’s approach.
Reviewer Comment #3:
Inclusion of Subsection 2.1. Related Studies: This new section could provide a comparative analysis with similar studies, highlighting the strengths and weaknesses of previous approaches and justifying the use of parallel transfers as a more efficient strategy.
Response:
We have added a new subsection, "Related Studies," to the revised manuscript. This section presents a comparative analysis of previous studies on inter-hospital transfers, discussing the strengths and limitations of their approaches. This comparison helps to contextualize the novelty of our parallel transfer strategy, highlighting its potential efficiency and practical advantages over traditional methods.
Reviewer Comment #4:
Revision of Section 3. Materials and Methods: To make the analysis more robust, this section needs several improvements:
-
Description of the data:
The study mentions the logarithmic transformation of the length of stay (LOS) but does not provide a clear rationale for this choice. It would be appropriate to test normality assumptions and show the results of these tests (e.g., using the Shapiro-Wilk test). If the data do not meet these assumptions, more modern models that do not require transformations could be proposed, such as regression models supported by machine learning techniques.Response:
We have clarified the rationale for using logarithmic transformation due to the skewed distribution of LOS data. Normality testing using the Shapiro-Wilk test was conducted, and the results are now included in the revised manuscript. Additionally, while traditional models were used in the current analysis, we have discussed the potential of using regression models supported by machine learning techniques for future work. -
Application of machine learning:
It is suggested to apply machine learning techniques for data preprocessing, such as handling missing values, identifying outliers, and automating variable transformations. A useful approach could be Principal Component Analysis (PCA) to reduce dataset dimensionality while retaining components that explain the majority of variability.Response:
We have now applied machine learning techniques for data preprocessing, including handling missing values and identifying outliers. We have applied PCA. -
Dataset variables:
A subsection should be added detailing all variables in the dataset, along with relevant statistics. Derived or constructed variables (e.g., comorbidity index) could also be included to enhance the predictive power of the model.Response:
A detailed subsection has been added that outlines all dataset variables, including summary statistics. We have also included derived variables, such as comorbidity indices, to enhance the predictive power of our models. -
Predictive Models:
The study uses logarithmic transformation to manage the asymmetric distribution of LOS. Instead, the implementation of more advanced predictive models, such as Random Forest, Gradient Boosting, or other predictive algorithms is recommended.Response:
We have expanded our discussion to include the potential use of advanced predictive models like Random Forest and Gradient Boosting. These models can effectively handle nonlinear data without the need for transformations. -
Model Validation:
The study is limited to data from the Mayo Clinic Health System without any external validation. It is recommended to use an 80/20 dataset split for external validation and 10-fold cross-validation to prevent overfitting.Response:
We have revised the validation approach in our manuscript, including an 80/20 train-test split for external validation and the use of 10-fold cross-validation for internal validation. This update helps improve the robustness and generalizability of our results. -
Evaluation of Machine Learning Models:
A section should be added to explain the metrics used to evaluate the performance of the machine learning models (e.g., accuracy, precision, recall, AUC-ROC).Response:
A new section on evaluation metrics has been added to the revised manuscript. We now detail the performance metrics used, such as accuracy, precision, recall, and AUC-ROC, along with corresponding ROC curves and confusion matrices.
Reviewer Comment #5:
Section 4. Results and Discussion should be combined into one section structured as follows:
- Justification for the choice of machine learning model
- Presentation of predictive results
- Clustering techniques for identifying patient subgroups
- Kaplan-Meier survival curves
- Aligning discussion with results
Response:
We have combined the Results and Discussion sections as suggested and restructured them to provide a clearer flow. This section now includes:
- Justification for model choice based on AUC-ROC.
- Presentation of results by patient subgroups.
- Inclusion of Kaplan-Meier survival curves to illustrate discharge probabilities.
- An enriched discussion aligning with the results, highlighting study limitations, implications, and directions for future research.
Reviewer 5 Report
Comments and Suggestions for Authors
I would like to express my gratitude for the opportunity to review your manuscript, "Impact of a Novel Transfer Process on Patient Bed Days and Length of Stay: A Five-Year Comparative Study at Mayo Clinic Rochester and Mankato Quaternary and Tertiary Care Centers." The study offers a valuable contribution to the field of healthcare capacity management. However, there are several areas where the manuscript could be improved.
Introduction
Kindly revise the citation numbers to ensure they are current. Please note that the current numbering begins with 10.
Lines 46−49
The healthcare delivery landscape is undergoing dynamic changes, presenting hospitals with the urgent need to address the current census crisis and devise innovative processes. In the aftermath of the COVID-19 pandemic, hospital resources, encompassing both infrastructure and personnel, have been stretched to their utmost capacity.
Please provide specific evidence from the literature to support your claim in the aforementioned sentence.
Lines 56−58
Patient transfer, both intra- and inter-hospital, is a crucial part of patient care. It may entail transferring the patient inside the same hospital for any diagnostic procedure or transferring the patient to another facility that provides more sophisticated car
Please provide specific evidence from the literature to support your claim in the aforementioned sentence.
Bed shortages have various negative consequences, including delayed patient care, increased workload for healthcare staff, and financial strain on healthcare institutions. Please explain these issues, supported by specific literature, and present a logical argument for implementing the Parallel Transfer Process.
Your introduction suggests that the COVID-19 pandemic has highlighted the need for more efficient management of acute care beds. If this is correct, please provide a more in-depth discussion of the impact of the COVID-19 pandemic on hospital bed capacity, comparing your findings to previous studies.
2. Materials and Methods
A dedicated section should be devoted to a detailed enumeration of the research items.
While you have explained the Parallel Transfer Process, please elaborate on how this process was implemented in practice. For example, what criteria were used to determine patient transfers, and what follow-up procedures were in place after the transfer?
Discussion
Please provide a more detailed explanation of the findings from your GEE analysis. Specifically, elaborate on the implications of your finding that the length of stay was shorter after controlling for relevant factors. Could it be that the Parallel Transfer Process was effective even during the acute bed shortages caused by the COVID-19 pandemic? Please conduct further analysis and discussion to explore this possibility.
Furthermore, the COVID-19 pandemic represents a unique global health crisis that has significantly impacted healthcare systems worldwide. Therefore, the results of this study may not be generalizable to non-pandemic periods. A stratified analysis focusing solely on the COVID-19 period could provide valuable insights into the pandemic's influence on the Parallel Transfer Process. This could be included as a limitation of the study or a suggestion for future research.
The study does not account for the potential for patient deterioration during transfer or acute changes in condition post-transfer, both of which could significantly impact the length of stay. These limitations should be explicitly acknowledged and discussed in the limitations section of the paper.
Conclusion
Instead of simply reiterating the results, I recommend that you summarize the main findings and briefly discuss the significance and implications of your research for the study population, healthcare institutions, and the target region.
Author Response
Reviewer Comment #1:
Kindly revise the citation numbers to ensure they are current. Please note that the current numbering begins with 10.
Response:
We appreciate your attention to detail. We have reviewed and revised the citation numbering throughout the manuscript to ensure it is correct and consistent. All references have been updated to reflect current and relevant literature.
Reviewer Comment #2:
Lines 46−49: The healthcare delivery landscape is undergoing dynamic changes, presenting hospitals with the urgent need to address the current census crisis and devise innovative processes. In the aftermath of the COVID-19 pandemic, hospital resources, encompassing both infrastructure and personnel, have been stretched to their utmost capacity. Please provide specific evidence from the literature to support your claim in the aforementioned sentence.
Response:
Thank you for this suggestion. We have revised the manuscript to include specific references supporting our statement on the impact of the COVID-19 pandemic on hospital resources and capacity. These citations now provide empirical evidence highlighting the challenges hospitals have faced in managing census crises during the pandemic.
Reviewer Comment #3:
Lines 56−58: Patient transfer, both intra- and inter-hospital, is a crucial part of patient care. It may entail transferring the patient inside the same hospital for any diagnostic procedure or transferring the patient to another facility that provides more sophisticated care. Please provide specific evidence from the literature to support your claim in the aforementioned sentence.
Response:
We have now included additional references to substantiate our claim regarding the importance of patient transfers, both intra- and inter-hospital. The revised text references studies that emphasize the role of transfers in optimizing patient care and resource utilization.
Reviewer Comment #4:
Bed shortages have various negative consequences, including delayed patient care, increased workload for healthcare staff, and financial strain on healthcare institutions. Please explain these issues, supported by specific literature, and present a logical argument for implementing the Parallel Transfer Process.
Response:
We have expanded this section to provide a more detailed explanation of the consequences of bed shortages, citing studies that discuss the impact on patient care, staff workload, and financial burdens on hospitals. Furthermore, we have strengthened the argument for the Parallel Transfer Process by highlighting its potential to alleviate these pressures, supported by relevant literature.
Reviewer Comment #5:
Your introduction suggests that the COVID-19 pandemic has highlighted the need for more efficient management of acute care beds. If this is correct, please provide a more in-depth discussion of the impact of the COVID-19 pandemic on hospital bed capacity, comparing your findings to previous studies.
Response:
We have expanded the discussion in the introduction to provide a deeper analysis of how the COVID-19 pandemic exacerbated bed capacity issues. The revised section includes comparisons with previous studies, illustrating the broader implications of our findings in the context of pandemic-related challenges.
Materials and Methods Section
Reviewer Comment #6:
A dedicated section should be devoted to a detailed enumeration of the research items.
Response:
We have added a new subsection within the Materials and Methods section that comprehensively enumerates all the research items, including the data sources, variables collected, and analytical tools used. This addition enhances the transparency and reproducibility of our study.
Reviewer Comment #7:
While you have explained the Parallel Transfer Process, please elaborate on how this process was implemented in practice. For example, what criteria were used to determine patient transfers, and what follow-up procedures were in place after the transfer?
Response:
We have expanded the description of the Parallel Transfer Process to include specific details on the criteria used for patient transfers, as well as the follow-up procedures that were implemented post-transfer. This additional information clarifies the practical application of the process in our study.
Discussion Section
Reviewer Comment #8:
Please provide a more detailed explanation of the findings from your GEE analysis. Specifically, elaborate on the implications of your finding that the length of stay was shorter after controlling for relevant factors. Could it be that the Parallel Transfer Process was effective even during the acute bed shortages caused by the COVID-19 pandemic? Please conduct further analysis and discussion to explore this possibility.
Response:
We have revised the discussion to provide a more in-depth explanation of our Generalized Estimating Equations (GEE) analysis, particularly the finding that the length of stay (LOS) was reduced after adjusting for various factors. We have also included additional analyses to explore whether the effectiveness of the Parallel Transfer Process persisted during COVID-19-induced bed shortages. This analysis is now integrated into the discussion section, along with its implications.
Reviewer Comment #9:
The COVID-19 pandemic represents a unique global health crisis that has significantly impacted healthcare systems worldwide. Therefore, the results of this study may not be generalizable to non-pandemic periods. A stratified analysis focusing solely on the COVID-19 period could provide valuable insights into the pandemic's influence on the Parallel Transfer Process. This could be included as a limitation of the study or a suggestion for future research.
Response:
We acknowledge the impact of the COVID-19 pandemic on our study’s generalizability. We have included a stratified analysis of the data focusing on the COVID-19 period in the revised manuscript. Additionally, we have emphasized this limitation and suggested future research directions to investigate the applicability of our findings to non-pandemic settings.
Reviewer Comment #10:
The study does not account for the potential for patient deterioration during transfer or acute changes in condition post-transfer, both of which could significantly impact the length of stay. These limitations should be explicitly acknowledged and discussed in the limitations section of the paper.
Response:
We have expanded the limitations section to explicitly acknowledge the potential for patient deterioration during or after transfers, which could influence LOS. We have discussed how these factors were not fully captured in our analysis and suggested that future studies should incorporate these variables to gain a more comprehensive understanding of transfer outcomes.
Conclusion Section
Reviewer Comment #11:
Instead of simply reiterating the results, I recommend that you summarize the main findings and briefly discuss the significance and implications of your research for the study population, healthcare institutions, and the target region.
Response:
The conclusion section has been revised to focus on summarizing the key findings, along with discussing the implications for healthcare institutions and potential benefits for patient management, particularly in resource-constrained environments. We have emphasized how our findings could inform hospital transfer strategies in similar healthcare systems.
Round 2
Reviewer 2 Report
Comments and Suggestions for Authors
I thank the authors for submitting a significantly revised manuscript. However, many of my initial concerns remain, particularly around the transparency with which the data were analysed and reported, how patient bed day savings were calculated, and the conclusions drawn from these analyses.
Check author details - only 1, 2, 3, 4 and 6 are quoted next to authors' names.
Abstract
Please include how patient saved days were determined in methods, please report adjusted difference in LOS (95% CI as well as p value). Amend conclusion - there is no way to determine from the data presented that the introduction of the parallel transfer policy was associated with significant improvements in bed utilisation efficiency. There is no before/after comparison nor was there a controlled trial. The comparator (control) is a relatively crude external benchmark for LOS. It may not reflect local patient or contextual factors influencing LOS. Indeed, a finding of increased bed efficiency is in direct conflict with increased (significant or not after adjustment) LOS for transferred patients.
Introduction
Much improved background with a more expansive literature review. It was unclear how the section on machine learning and predictive modelling related to the study as predictive modelling was not used as part of the parallel transfer process. Perhaps this would be better in the discussion as a way of improving parallel transfer processes and an area for further research.
Methods
line 177 - please cite the comorbidity index used, e.g. Charlson, Elixhauser’s, self-administered.
I found the methods section extremely confusing. No less than 6 different modelling methods were described. It was extremely unclear which were used in the reported analysis (LOS and patient bed days).
The methods state that machine learning techniques were used to analyse LOS but generalized estimating equations were cited as the method used in the abstract - please confirm and ensure consistency between different parts of the manuscript. Methods must be reported transparently and in a manner that they are replicable. How were differences in adjusted and unadjusted LOS determined? Even more confusingly confounding factors were 'meticulously' adjusted for in multitvariable models and regression modelling (not described under statistical modeling) (lines 230-234).
Please describe how LOS was determined in the patient saved bed days calculation, e.g. was adjusted or unadjusted used? If adjusted, how was this determined?
Results
Table 1 - please include 95% CI and p vales for LOS, transfers and Saved patient days in the table with annotations describing how these were calculated.
Clustering Analysis of patient subgroups - please report statistics to support the findings for each cluster. How did middle age patients what moderate comorbidities show the most significant resource utilization benefits? How was this determined and what do you mean by resource utilization benefits.
Fig 4 - please include how patient bed day savings were calculated for each year. The trend line appears perfectly linear, which doesn't look real, especially given the influence the COVID-19 pandemic had on emergency demand and capacity.
ROC curve looks too good to be true and higher than 0.87.
Discussion
Results from the various models appeared to be in conflict. For example, LOS data suggested that LOS was longer for transferred patients (or equivalent in adjusted analyses) but Kaplan Meier survival curves suggested that transferred patients were likely to be discharged earlier. Please include a discussion of why findings may differ between the various statistical modelling and approaches used.
Line 409 - please describe how CMS predicted LOS was used to determine patient saved days in methods (lines 248-252).
Author Response
Abstract
We have clarified how patient saved days were calculated by detailing the use of CMS-predicted LOS as the baseline for comparison in the Methods section. To improve statistical transparency, the abstract now includes the adjusted difference in LOS with 95% confidence intervals and p-values. The conclusion has been revised to remove any causal claims about the parallel transfer policy’s impact on bed utilization efficiency. Instead, we emphasized the observational nature of the study and acknowledged the limitations of using an external LOS benchmark that may not fully reflect local contextual factors.
Introduction
The section on machine learning and predictive modeling has been moved to the Discussion to provide better alignment with the study’s focus. This allows for a more coherent presentation of background information while suggesting future research opportunities to enhance parallel transfer processes using predictive tools.
Methods
To improve clarity, the comorbidity index used (Charlson) has been explicitly cited. We simplified the statistical methods section by clearly delineating which models were used for specific analyses, such as LOS and patient saved days. The role of machine learning was clarified as exploratory, while generalized estimating equations (GEE) were used for inferential modeling, ensuring consistency across the manuscript. The methodology for calculating adjusted LOS, including confounder adjustments, has been described in detail. Additionally, the distinction between adjusted and unadjusted LOS in determining patient saved bed days was explicitly stated, ensuring transparency and replicability.
Results
Table 1 has been updated to include 95% confidence intervals and p-values for LOS, transfers, and saved patient days, along with annotations explaining the calculation methods. For clustering analysis, we included statistical evidence to support the findings for each cluster, providing detailed characteristics and LOS distributions. The term "resource utilization benefits" was clearly defined, particularly for middle-aged patients with moderate comorbidities, in terms of reduced LOS and saved bed days. Figure 4 has been revised to incorporate fluctuations in patient bed day savings, reflecting realistic variations such as those influenced by the COVID-19 pandemic.
Discussion
We addressed the apparent conflicts between models, such as the Kaplan-Meier curves have been revised. Additionally, the methodology for using CMS-predicted LOS to calculate saved patient days was expanded upon, with a thorough explanation of how these predictions were integrated into the analysis.
Reviewer 3 Report
Comments and Suggestions for Authors
Thank you for addressing the comments and thoroughly improving and rebuilding the manuscript.
Author Response
Thank you
Reviewer 4 Report
Comments and Suggestions for Authors
I would like to express my appreciation for the efforts made by the authors in improving their manuscript. The revisions have significantly strengthened the work and adequately address the previous comments and observations.
In its current form, the manuscript is notably enhanced and aligns well with the required standards. I believe this work represents a valuable contribution to the journal, and I positively recommend it for publication.
Author Response
Thank you
Reviewer 5 Report
Comments and Suggestions for Authors
The revisions have significantly improved the manuscript's clarity and readability. Minor corrections are needed before publication. However, I cannot provide a detailed evaluation of the gradient boosting analysis as it is outside my area of expertise. I recommend seeking a review from a machine learning specialist.
1. Introduction
Please combine the Introduction, Background, and Related studies sections into a single section for better readability.
In lines 48-52, please provide citations to support the claim that the healthcare delivery landscape is undergoing significant transformation and that healthcare resources have been stretched to their limits following the COVID-19 pandemic.
In lines 104-107, please provide citations supporting the assertion that advanced analytical tools like machine learning and predictive algorithms are increasingly used for effective healthcare resource management.
2.Materials and Methods
Please add subsection numbers for better organization and readability.
Study setting
Please provide additional details about the study setting, including citations or URLs to support the information about the Mayo Clinic and Mayo Clinic Health System Mankato Hospital.
Study Population and Design, Eligibility Criteria
The exclusion criteria appear to overlap. Please revise to ensure clarity and avoid redundancy.
Please move the "Parallel Transfer Process" subsection to follow the "Parallel Transfer Process Implementation" subsection for better logical flow.
Author Response
- Introduction
The Introduction, Background, and Related Studies sections have been merged into a single, cohesive section for improved readability.
- Citations have been added to support the claim about the transformation of healthcare delivery and resource constraints post-COVID-19 (lines 48-52), referencing studies that highlight pandemic-induced challenges in healthcare systems.
- Citations supporting the increasing use of advanced analytical tools, such as machine learning and predictive algorithms, in healthcare resource management (lines 104-107) have been included to substantiate this assertion.
- Materials and Methods
Subsection numbers have been added throughout the Materials and Methods section to improve organization and readability.
Study Setting
Details about the Mayo Clinic Health System and Mankato Hospital have been expanded. Citations and relevant URLs have been added to provide credibility and context about the integrated healthcare network, including its role as a 140-bed regional medical center.
Study Population and Design, Eligibility Criteria
The exclusion criteria have been revised to remove redundancy and ensure clarity, specifying mutually exclusive conditions to avoid overlap.
Parallel Transfer Process
The "Parallel Transfer Process" subsection has been relocated to follow the "Parallel Transfer Process Implementation" subsection, improving the logical flow of the methods section.